# DCT Domain Detail Image Enhancement for More Resolved Images

**Seongbae Bang** [1] and **Wonha Kim** [2,*]

1 Department of Electronic Engineering, Kyung Hee University, Yongin-si 17103, Gyeonggi-do, Korea; sungbae9023@hanmail.net
2 Department of Electronic Engineering (AgeTech-Service Convergence Major), Kyung Hee University, Yongin-si 17103, Gyeonggi-do, Korea
* Correspondence: wonha@khu.ac.kr; Tel.: +82-31-201-2030

**Abstract:** This paper develops a detail image signal enhancement that makes images perceived as being clearer and more resolved and so more effective for higher resolution displays. We observe that the local variant signal enhancement makes images more vivid, and the more revealed granular signals harmonically embedded on the local variant signals make images more resolved. Based on this observation, we develop a method that not only emphasizes the local variant signals by scaling up the frequency energy in accordance with human visual perception, but also strengthens the granular signals by embedding the alpha-rooting enhanced frequency components. The proposed energy scaling method emphasizes the detail signals in texture images and rarely boosts noisy signals in plain images. In addition, to avoid the local ringing artifact, the proposed method adjusts the enhancement direction to be parallel to the underlying image signal direction. It was verified through subjective and objective quality evaluations that the developed method makes images perceived as clearer and highly resolved.

**Keywords:** image detail enhancement; DCT domain perceived contrast; perceptual image quality

## 1. Introduction

Humans recognize more sharpened images as being clearer and perceive images embedding finely resolved signals as being higher resolution images, even at the same resolution. So, as image contents are increasingly produced toward higher quality and are presented at higher resolution displays, state-of-the-art detail image enhancements need to make images clearer and finely resolved.

Figure 1 shows a comparison of the original image with one that is clearer and another that is clearer and more finely resolved. To get a better understanding, one-dimensional horizontal signals of the images are plotted. The image in Figure 1b is clearer than the original image because magnified local variations sharpen the image. In the image in Figure 1c, the local variations are magnified, and, simultaneously, the granular variations indicated by circles are embedded while preserving the contour of the local variant signals. The granular signals are the most resolved signals for an image display to represent. Owing to the granular variations, humans tend to perceive the image shown in Figure 1c as being the highest resolution.

Since existing detail enhancement methods increase local contrast only without observing the different effects of local variants and granular signals, they enhance only the sharpness of the images. Therefore, it is necessary to develop a detail image enhancement capable of magnifying local variations and simultaneously emphasizing granular signals in harmony with local variant signals.

The detail enhancement methods can be roughly categorized into spatial domain, frequency domain, and learning-based methods. Spatial domain methods focus primarily on elevating local variant signals. In the spatial domain, Majumder et al. [1] intensified

the local contrasts based on Weber's law. Kou et al. equalized the overall gradient by increasing the image signal gradients at small local variant regions [2]. Deng et al. applied a multi-resolution filter to decompose image signals into base and detail image signals and then emphasized the detail image signals [3]. Nercessian et al. measured the energy ratio among different resolution signals in the wavelet transform domain and then increased the energy ratio of high-resolution signals [4]. Since the granular signals are tiny compared to local variant signals, these methods are technically unable to extract the granular signals and thus rarely enhance them. Therefore, although the spatial domain methods usually enhance the local contrast for the images to become clearer, they rarely enhance images to be finely resolved.

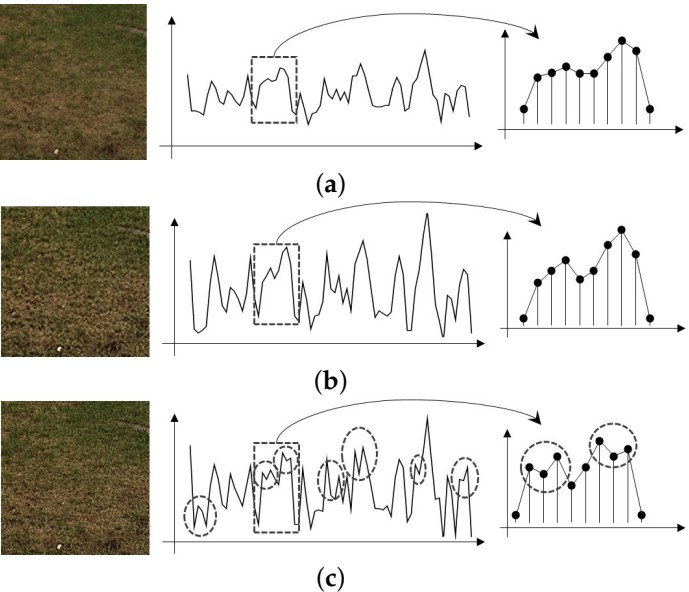

**Figure 1.** Effects of detail image signals. (**a**) Original image. (**b**) Image with increased local variants. (**c**) Image with increased local variants and enhanced granular signals.

Using the capability of dissolving the image signals into frequency components, the frequency domain methods have focused on how to increase the frequency energy of detail signals. The multi-band energy scaling method (MESM) developed by Tang et al. recursively scales up the frequency energy ratio in the discrete cosine transform (DCT) domain as the frequency band increases [5]. This method sharpens local variant signals in line with human visual perception, but rarely makes images seemed more resolved. The alpha rooting method exponentially boosts the frequency energy inversely to the original frequency energy [6] and results in the granular signals well, but often produces noisy signals. Celik conducted the DCT over the entire image to utilize an extremely fine frequency resolution [7]. This method weights higher frequency components as being greater in proportion to the global variation. Since the method processes detail signals globally, it may produce insufficiently enhanced textures or excessively boosted noises. Moreover, it is not easy in actual systems to take the DCT over the entire image. Therefore, the existing frequency domain methods either rarely reveal granular signals or may produce noisy signals when they reveal granular signals.

More recently, learning-based image-enhancement methods have been developed and are effective for improving global contrast or tuning color tones, such as de-hazing, low-light and underwater areas [8,9]. Yan et al. applied a convolution neural network (CNN) that was learned from images enhanced by algorithms or human experts [10]. Gharbi et al. designed a bilateral CNN separately learning global and local variant signals to achieve real-time processing, even on mobile devices [11]. Chen et al. proposed a GAN-based image enhancement network that overcomes the ill-convergence that often occurs in GAN [12]. Since the GAN-based methods are unsupervised approaches, the method inherently bears

the possibility of producing unnatural image signals. Because learning-based methods could easily lose detailed signal information in deep-hidden layers, the methods may generate insufficient detail signals or erroneous detail signals. Therefore, learning-based methods mainly improve the global contrast and color brightness; however, it has limitations in generating or inferring detail image signals. Moreover, learning-based methods commonly require heavy computation, compared to model-based methods.

We develop a frequency domain method that enhances an image to be perceived as being both clearer and of a higher resolution, distinguished from existing methods that enhance images to be clearer, only. The proposed method further decomposes the detail image signals into local variant and granular signals. To increase the sharpness of local variant signals, we devise a recursive frequency energy scaling-up method from the perceived contrast model that indicates the visual sensitivity of detail signals in the frequency domain. We enhance the frequency components by the alpha-rooting while scaling up the frequency energy to embed the granular signals harmonically on the local variant signals. We also design the energy scalar to emphasize the detail image signals at texture images and suppress the increasing noisy patterns in plain images. Additionally, to reduce the ringing artifact, we devise a method for tuning the enhancement direction to be parallel with the signal direction in the DCT domain.

The remainder of this paper is organized as follows. Section 2 discusses the perceived contrast measure in the DCT domain. Section 3 proposes the perceptual contrast increment method that recursively modifies DCT coefficients and presents a method to avoid artifacts and noise boosting. Section 4 evaluates the proposed method's performance, compared to existing enhancement methods, and analyzes the artifacts caused by enhancement methods. Section 5 reaches conclusions.

## 2. DCT Domain Human Perceptual Contrast

Many psychological and physiological studies have reported that human visual neurons accept visual signals in frequency components; thus, human visual perception is primarily affected by the frequency energy distributions of images [13,14]. The image signal components in the frequency domain are also efficiently separated and robustly processed. Therefore, we adopt the DCT as the enhancement platform.

Let $f(i, j)$ be an image pixel value at position $(i, j)$ of the $N \times N$ DCT block. The DCT coefficient $F(u, v)$ is obtained as follows:

$$F(u, v) = C_u C_v \sum_{i=0}^{N-1} \sum_{j=0}^{N-1} f(i, j) \cdot \cos \frac{(2i + 1)u\pi}{2N} \\ \cdot \cos \frac{(2j + 1)v\pi}{2N}, \quad 0 \leq u, v \leq N - 1 \tag{1}$$

where

$$C_p = \begin{cases} \sqrt{1/N} & \text{for} \quad p = 0, \\ \sqrt{2/N} & \text{otherwise.} \end{cases}$$

When the ratio between the viewing distance and display height is $R_d$ and the vertical pixel number of the displayed image is Pix, the spatial frequency, $\omega(u, v)$, in actual viewing conditions is converted into the DCT frequency as follows:

$$\omega(u, v) = \frac{1}{2N\theta} \sqrt{u^2 + v^2} \tag{2}$$

where

$$\theta = 2 \cdot \arctan\left(\frac{1}{2 \cdot R_d \cdot \text{Pix}}\right).$$

Several studies have found that the human physiological visual sensitivity varies in the spatial frequency, is highest at $3 \sim 5$ cycles/degree, and is higher in the vertical and horizontal directions than in the diagonal direction because of the oblique effect. The

studies also have modeled the visual sensitivity in the DCT domain, referred to as the contrast sensitivity function (CSF) [15,16]. The CSF in the DCT domain is as follows:

$$
\begin{aligned}
CSF(u,v) \;\; &= \text{Frequency Sensitivity} \times \text{Oblique Effect} \\
&= 0.25 \cdot \left[ \frac{1}{\phi_u \phi_v} \cdot \frac{\exp(0.18 \cdot \omega(u,v))}{1.33 + 0.11 \cdot \omega(u,v)} \right] \\
&\quad \cdot \left[ \frac{1}{0.6 + 0.4\cos(\psi(u,v))^2} \right]
\end{aligned}
\tag{3}
$$

where the direction angle is $\psi(u,v) = \arcsin(2\omega(u,0)\omega(0,v)\,/\,\omega(u,v)^2)$.

In addition to the CSF, the human visual sensitivity at specific frequency is also affected by the frequency energy distribution of an underlying image. Haun and Peli conducted experiments measuring human visual sensitivities to stimuli with different frequencies and directions, deriving a human visual sensitivity model, called perceptual contrasts (PC) [13,14]. The perceptual contrasts (PC) are as follows:

$$
\text{PC}(u,v) = \frac{|F(u,v)|^{2.4}}{CSF(u,v)^{-2} + B(u,v) + |F(u,v)|^2}.
\tag{4}
$$

where $B(u,v)$ is the background energy at $(u,v)$. The background energy with respect to $F(u,v)$ is the energy accumulation of the frequency components lower than $(u,v)$ [5,14]. That is,

$$
B(u,v) = \sum_{p=0}^{u-1} \sum_{q=0}^{v-1} |F(p,q)|^2.
\tag{5}
$$

The PC indicates that human visual sensitivity is higher for the frequency components with a larger CSF value, lower background energy and larger frequency energy [13,14]. While the existing contrast measurements quantify image signal variations in the spatial domain, the PC measures how much a human actually perceives each frequency component and thus, provides the contrast measure more matched with human visual perception.

## 3. Development of Human Perception Oriented Detail Image Enhancement

We develop a detail image signal enhancement method that recursively increases the perceptual contrast (PC) and simultaneously intensifies the granular signals. In addition, to avoid the ringing artifact, we devise a method that adjusts the enhancement direction to be parallel to the image signal direction.

### 3.1. Perceptual Contrast (PC) Based Energy Scaling Method

Human visual perception generally prefers images with higher visual sensitivity [17]. In order to increase human perceived visual sensitivities at frequency components, we propose a method that recursively scales up the PC as the frequency proceeds from low to high band.

The original and enhanced DCT coefficients are denoted as $F(u,v)$ and $\bar{F}(u,v)$, respectively. Subsequently, the perceptual contrast at $(u,v)$ of the original and enhanced images are denoted as $\text{PC}(u,v)$ and $\overline{\text{PC}}(u,v)$, respectively. Introducing the PC-enhancing scalar, $\lambda(>1)$, $\overline{\text{PC}}(u,v)$ is related to $\text{PC}(u,v)$ in the following way:

$$
\overline{\text{PC}}(u,v) = \lambda \cdot \text{PC}(u,v), \quad 0 \le u,v \le (N-1).
\tag{6}
$$

By inserting (4) to (6), the following equation is obtained:

$$
\bar{F}(u,v) = \lambda \cdot R(u,v) \cdot F(u,v)
\tag{7}
$$

where $R(u,v)$ is the energy scaling factor and the following holds:

$$R(u,v) = \left\{ \frac{CSF(u,v)^{-2} + \bar{B}(u,v) + |\bar{F}(u,v)|^2}{CSF(u,v)^{-2} + B(u,v) + |F(u,v)|^2} \right\}^{\frac{1}{2.4}}. \tag{8}$$

To enhance the granular signals, the alpha rooting method is exploited, which emphasizes the energy of high-frequency components. The frequency component enhanced by the alpha rooting method is as follows:

$$\bar{F}(u,v) = \left| \frac{F(u,v)}{F(0,0)} \right|^{\alpha-1} \cdot F(u,v), \quad 0 < \alpha < 1 \tag{9}$$

where $\alpha$ is the enhancement factor. As $\alpha$ is closer to 0, the higher-frequency components are emphasized to generate more granular signals.

Then, $R(u,v)$ embedding the alpha routine enhancement becomes the following:

$$R(u,v) = \left\{ \frac{CSF(u,v)^{-2} + \bar{B}(u,v) + \left| \frac{F(u,v)^{\alpha}}{F(0,0)^{\alpha-1}} \right|^2}{CSF(u,v)^{-2} + B(u,v) + |F(u,v)|^2} \right\}^{\frac{1}{2.4}} \tag{10}$$

$R(u,v)$ is recursively updated from (7) and (10) as the frequency increases.

The role of $R(u,v)$ is to control the enhancement in adaptation with the characteristics of the image signals. Since $CSF(u,v)$ has high values at middle-frequency components corresponding to local variant image signals, $R(u,v)$ correspondingly has large values at these frequency bands so that it primarily enhances the local variant signals to produce sharpened images. Because the alpha rooting enhanced frequency component is embedded into $R(u,v)$, the granular signals become more visible, while the noise signals probably occurring from the enhanced high-frequency signals are prevented. Thus, in texture images typically having a large energy at middle- and high-frequency components, $R(u,v)$ enhances the detail signals while revealing the granular signals. In plain images that do not contain many detail image signals, the frequency energies of detail image signals are much smaller that those of $CSF(u,v)$, and $CSF(u,v)$ dominates over the background energy. Therefore, $R(u,v)$ becomes approximately 1 at all frequency bands and rarely produces noise signals that may occur if the plain image signals are enhanced.

Figure 2 compares $R(u,v)$ for the texture and plain images and shows their detail enhanced images. The texture image is shown in Figure 2a, where the proposed method not only increases local variations but also embeds granular signals in local variant signals. The plane image is shown in Figure 2b, where the proposed method rarely enhances the image signals and does not produce noisy image signals.

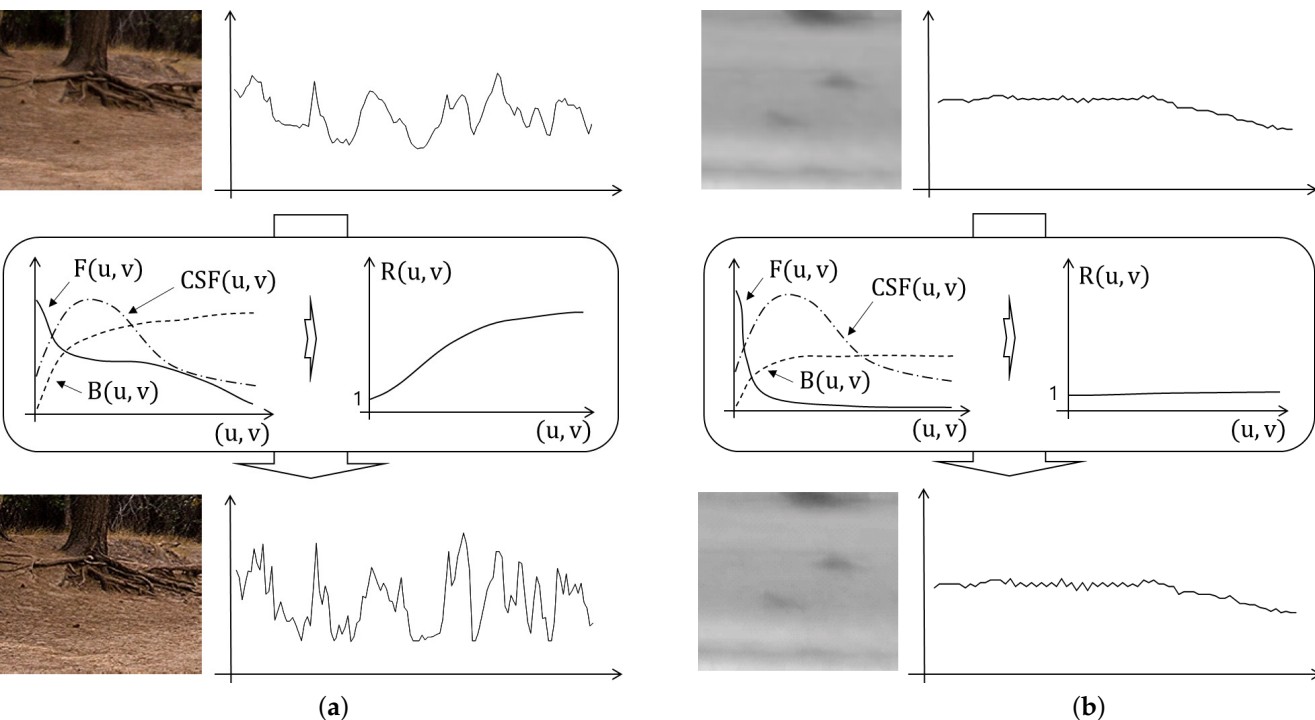

**Figure 2.** The proposed energy scalar spectrums for texture and plane images and their images enhanced by the proposed method. (**a**) A texture image. (**b**) A plane image.

### 3.2. Signal Direction Adaptive Enhancement

We develop the DCT domain method to remove the ringing artifact. When directional images, such as edges, are not processed in parallel with their directions, overshooting occurs, which appears as ringing patterns. The ringing artifact is apparent as the enhancement direction becomes more perpendicular to the image direction. To prevent ringing artifacts, the enhancement direction must be parallel to the signal direction.

Figure 3 shows the DCT energy distribution of edge and texture images. As shown in Figure 3a, when an image signal is directed more vertically, the energy of the DCT coefficients is further condensed in the first row. Conversely, more horizontally directed signals have greater energy from the DCT coefficients in the first column as shown in Figure 3b. In the diagonal signals, the DCT coefficients are symmetric such as in Figure 3c. Therefore, the magnitudes of the first column and row DCT coefficients are equivalent to the gradients in the vertical and horizontal directions [18,19]. Let $\nabla_{ver}$ and $\nabla_{hor}$ be the vertical and horizontal gradients, respectively. Then, the following hold:

$$\nabla_{ver} = \frac{1}{\Gamma} \sum_{u=1}^{N-1} |F(u,0)|, \quad \nabla_{hor} = \frac{1}{\Gamma} \sum_{v=1}^{N-1} |F(0,v)| \tag{11}$$

where

$$\Gamma = \sum_{u=0}^{N-1} \sum_{v=0}^{N-1} |F(u,v)| - |F(0,0)|.$$

As an image signal slants closer to the vertical direction, $\nabla_{ver}$ is larger than $\nabla_{hor}$. When an image signal is directed diagonally, $\nabla_{ver} = \nabla_{hor}$.

To adjust the energy scaling direction, the DCT coefficients are decomposed into horizontal and vertical directions. The recursive PC scaling factors for the horizontal and vertical directions are obtained as follows:

$$R_{hor}(v) = \sum_{u=0}^{N-1} R(u,v), \quad R_{ver}(u) = \sum_{v=0}^{N-1} R(u,v). \tag{12}$$

Then, the enhanced DCT coefficients for each direction become the following:

$$\bar{F}_{hor}(u,v) = \lambda \cdot R_{ver}(u) \cdot F(u,v)$$
$$\bar{F}_{ver}(u,v) = \lambda \cdot R_{hor}(v) \cdot F(u,v).$$

$$(13)$$

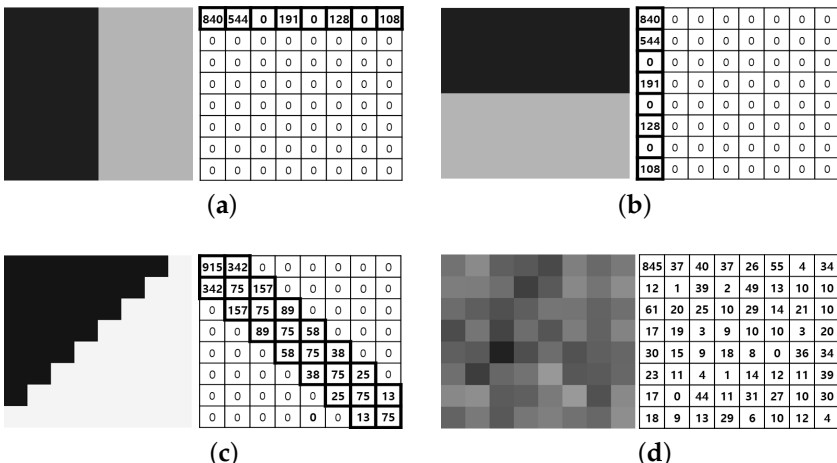

**(a)**

**(b)**

**(c)**

**(d)**

**Figure 3.** DCT energy distribution of edge and texture images. (**a**) DCT energy distribution of vertical edge. (**b**) DCT energy distribution of horizontal edge. (**c**) DCT energy distribution of diagonal edge. (**d**) DCT energy distribution of non-directional texture.

To steer the PC scaling direction in parallel with the image signal direction, we weight each gradient to vertical and horizontal enhanced DCT coefficients. So, we propose the direction adaptive enhanced DCT coefficients as the following:

$$\bar{F}(u,v)$$
$$= \frac{\nabla_{ver}}{\nabla} \cdot \bar{F}_{ver}(u,v) + \frac{\nabla_{hor}}{\nabla} \cdot \bar{F}_{hor}(u,v)$$
$$= \frac{\lambda}{\nabla} \cdot \{\nabla_{ver} \cdot R_{ver}(u) + \nabla_{hor} \cdot R_{hor}(v)\} \cdot F(u,v)$$

$$(14)$$

where $\nabla = \nabla_{ver} + \nabla_{hor}$.

The energies of the DC and low-frequency bands control the overall brightness of the DCT block. A change in the energy in the DC and low-frequency bands induces a brightness discontinuity among adjacent blocks, called the block artifact. The existing methods that avoid block artifacts do not scale the energies at DC and the first three frequency bands [20]. Following the existing methods, we do not scale the energy at the first three bands by setting $\lambda$ as follows:

$$\lambda = \begin{cases} 1, & \text{for } u + v \leq \lfloor N/3 \rfloor, \\ > 1, & \text{otherwise} \end{cases}$$

$$(15)$$

Figure 4 shows the edge images enhanced by the proposed method without the signal direction adaptive method and the proposed method with the signal direction adaptive method. Not using the signal direction adaptive method creates ringing artifacts at the sail edges; however, the direction-adaptive method tunes the DCT-coefficient scaling direction in parallel with the sail edge direction to avoid visible ringing artifacts.

Figure 5 shows the edge images enhanced by MESM, CWM and the proposed method. The proposed method and CWM do not create the ringing artifact, whereas MESM does, as it enhances image signals in all directions. The CWM linearly scales up the frequency components as the frequency increases. Hence, it does not significantly change low- and middle-frequency components corresponding to edge signals and avoids visible ringing

artifacts. As discussed in this section, the proposed method scales up the frequency components in parallel with the edge direction, and so does not make the ringing artifact.

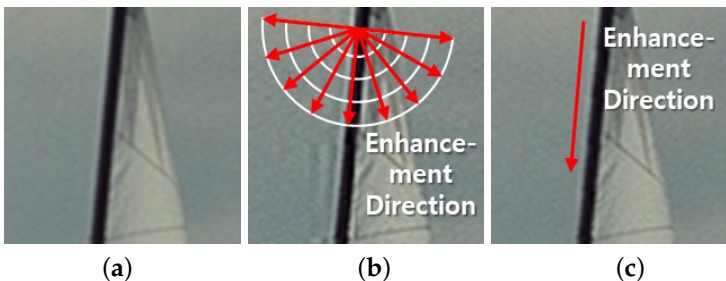

**Figure 4.** Reducing ringing artifact by direction-adaptive method. (**a**) Original edge image. (**b**) Image enhanced by the proposed method without direction-adaptive method. (**c**) Image enhanced using the proposed method with direction-adaptive method.

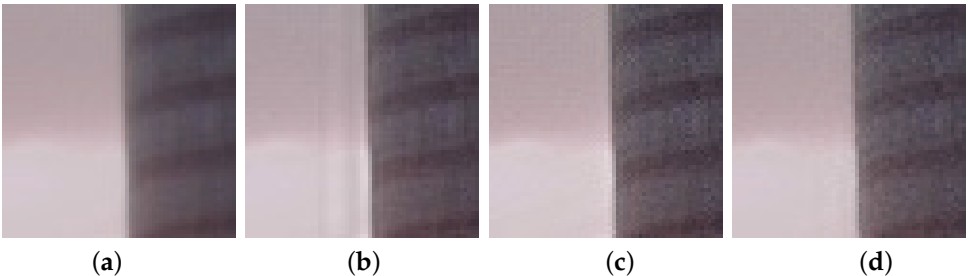

**Figure 5.** Edge image enhancement. (**a**) The original edge image. (**b**) The image enhanced by MESM. (**c**) The image enhanced by CWM. (**d**) The image enhanced by the proposed method.

### 3.3. Outline of the Proposed Method

Figure 6 shows an overview of the proposed detail image enhancement method. The method consists of detail signal enhancement and artifact reduction.

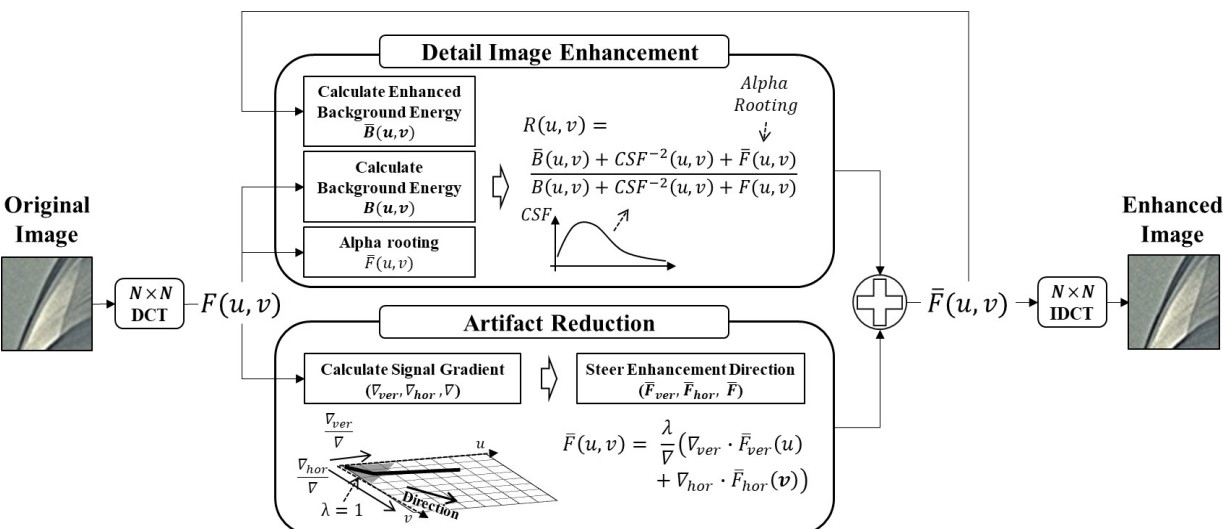

**Figure 6.** Overview of the proposed detail image signal enhancement.

In part of the detail image signal enhancement, the frequency energies are recursively scaled up in match with the human-perceived contrast. The proposed method designs the energy scalar $R(u,v)$ from the perceived contrast embedding the alpha-rooting enhanced frequency components. $R(u,v)$ enhances the detail image signals in texture images suppressing the enhancement of noisy signals in plain images. Additionally, since the

alpha rooting enhanced frequency component is embedded into the perceived contrast, the granular signals become more suitable for human visual perception.

In part of the artifact reduction, the proposed method measures the vertical and horizontal gradients in the DCT domain. To reduce ringing artifacts, the proposed method adjusts the frequency energy scaling direction parallel to the image signal direction by weighting each gradient to the enhanced DCT coefficients in the horizontal and vertical directions.

We analyze the computing complexity of the proposed method. Our method is the $O(n/N^2)$ time method, where $N$ is the block size. So, the large block size requires less computing time. If the block size is 16, the computing time of the proposed method is 118 msec per 1 mega pixel on a processor of intel I5-2500 3.3 GHz and 8 GB RAM. Thus, our proposed method would be processed in real time at full HD image.

## 4. Experiment and Discussion

We evaluate and analyze the performances of the proposed method in terms of objective and subjective image quality evaluations and artifact occurrence analysis. The test images are the ultra HD images in [21].

### 4.1. Objective and Subjective Image Quality Evaluation

The performance of the proposed method is evaluated and compared to recently developed local contrast enhancement methods, including unsharp masking [3], content adaptive image detail enhancement (CAIDE) [2], the multiband energy scaling method (MESM) [20] and the coefficient weight method (CWD) [7]. The unsharp masking method only enhances the detail signals after decomposing the image signals into the detail and base signals. The CAIDE exploits global optimization to enhance the detail signals by minimizing the less gradient region. By measuring the local contrasts with the energy ratio of the frequency bands, the MESM recursively scales the DCT coefficients to increase the measured contrasts. The CWM conducts the DCT over the entire image and linearly scales up the higher frequency coefficients. The unsharp masking and CAIDE are the spatial domain methods, and the proposed method, MESM and CWM are the frequency domain methods.

In objective quality evaluation, there are image signal feature–based and learning-based methods [22–25]. As an image signal feature–based method, the cumulative probability of blur detection (CPBD) metric is adopted, which has been reported to produce a more stable image quality score [22]. CPBD, a no-reference image quality assessment metric, measures the degree of artifacts, such as blurring, ringing and sharpness collectively and determines the numerical value indicating image quality. As the metric score approaches 1, the image quality is perceived to be better for humans. As a deep learning–based method, the full reference deep image quality assessment network (DeepIQA) [25] is used. DeepIQA allows a neural network to train the subjective image quality for predicting the differential mean opinion score (DMOS) values of the original and enhanced image pairs. The DeepIQA well evaluates image qualities in accordance with human visual perception because the DeepIQA exploits perceptual sensitivity map based on human visual perception. Positive DeepIQA values indicate the degree of quality improvement of the processed images over the original images.

For the subjective evaluation, we followed the categorical stimulus comparison judgment method recommended by ITU-R BT.500-11 [26]. We placed two identical Ultra HD monitors next to each other in ambient illumination of approximately 200 lux. The monitor resolution was set to 3840 × 2160 with the DPI at approximately 130. The color depth was 24 bits per pixel. The monitor luminance (brightness) was set at 250 cd/m$^2$, and the contrast ratio was 3000:1. For the 20/20 eyesight to fully observe the enhanced detailed signals, we set the viewing distance at about 1.5 m of which the vertical and horizontal viewing angles were about approximately 30° and 50°, respectively. [27] A total of 20 subjects were invited to compare the qualities of the 20 original images and images enhanced by the proposed method and the existing methods. The images were shown adjacent to each other

to guarantee blindness. Subjects assigned a score in the range of $[-3, 3]$ to the sequences. A score of $-3$ indicates that the left sequence has a significantly better visual quality than the sequence on the right whilst a score of 3 signifies that the sequence on the right has significantly better visual quality than the sequence on the left. A score of 0 indicates that no difference in visual quality is perceived.

In order to statistically analyze the improvement achieved by the proposed method, we conducted the Student's *t*-test [28]. The null hypothesis is that the proposed method or the conventional methods show no improvement, compared to the original images. We set the confidence level to be 95%; subsequently, the rejection region (rr) is $2.093\sigma/\sqrt{N}$, where $\mu$, $\sigma$, and $N$ are the average score, the standard deviation, and the number of subjects, respectively. The performance improvement scores of DMOS with 95% confidence can be calculated as $(\mu - rr)$.

Table 1 lists the scores of each method. It should be noted that CPBD and DeepIQA are objective measures, and DMOS is the subjective measure. The negative DMOS indicates that the subjects perceive the enhanced images as being of lower quality than the original images. Negative values usually occur when an enhancement method generates artifacts. The scores, even if expressed on different scales, show similar patterns, confirming the reliability of the experimental results. Because human perception is more precisely described in the frequency domain, the performance of frequency domain methods is usually higher than that of spatial domain methods. Among the frequency domain methods, the proposed method consistently produces competitive performances for most test images. In particular, the proposed method produced higher values in images containing more detail image signals.

**Table 1.** Image quality evaluation scores.

| | Method | Aerial | Bar | Boat | Cross Walk | Market | Nar-Tator | Square | Tango | Foun-Tain | Tunnel | Flag1 | Flag2 | Avg. |
|---|---|---|---|---|---|---|---|---|---|---|---|---|---|---|
| CPBD | Unsharp [3] | 0.54 | 0.78 | 0.36 | 0.72 | 0.65 | 0.48 | 0.55 | 0.80 | 0.50 | 0.60 | 0.91 | 0.46 | 0.61 |
| | CAIDE [2] | 0.54 | 0.76 | 0.45 | 0.78 | 0.70 | 0.74 | 0.60 | 0.78 | 0.53 | 0.69 | 0.82 | 0.61 | 0.67 |
| | MESM [20] | 0.57 | 0.81 | 0.41 | 0.74 | 0.68 | 0.63 | 0.62 | 0.81 | 0.56 | 0.62 | 0.92 | 0.55 | 0.66 |
| | CWM [7] | 0.59 | 0.80 | 0.49 | 0.80 | 0.77 | 0.70 | 0.66 | 0.84 | 0.59 | 0.69 | 0.92 | 0.65 | 0.71 |
| | Proposed | 0.69 | 0.81 | 0.61 | 0.85 | 0.81 | 0.67 | 0.69 | 0.86 | 0.68 | 0.74 | 0.93 | 0.73 | 0.76 |
| DIQA | Unsharp [3] | 0.12 | 0.11 | 0.08 | 0.08 | 0.15 | 0.11 | 0.07 | 0.09 | 0.07 | 0.08 | 0.12 | 0.06 | 0.10 |
| | CAIDE [2] | 0.07 | 0.05 | 0.13 | 0.08 | 0.12 | 0.10 | 0.17 | 0.11 | 0.06 | 0.07 | 0.11 | 0.02 | 0.09 |
| | MESM [20] | 0.15 | 0.14 | 0.12 | 0.19 | 0.19 | 0.19 | 0.18 | 0.14 | 0.12 | 0.12 | 0.17 | 0.19 | 0.16 |
| | CWM [7] | 0.30 | 0.29 | 0.22 | 0.31 | 0.37 | 0.31 | 0.26 | 0.20 | 0.21 | 0.26 | 0.30 | 0.26 | 0.27 |
| | Proposed | 0.31 | 0.29 | 0.19 | 0.31 | 0.37 | 0.31 | 0.23 | 0.22 | 0.20 | 0.23 | 0.30 | 0.24 | 0.27 |
| DMOS | Unsharp [3] | 0.15 | 0.06 | 0.25 | 0.14 | 0.13 | 0.15 | 0.04 | 0.06 | 0.23 | 0.04 | 0.18 | 0.13 | 0.13 |
| | CAIDE [2] | −0.85 | −0.21 | −0.93 | −0.60 | −0.21 | −0.63 | −0.81 | −1.18 | −0.67 | −0.18 | −0.31 | −0.15 | −0.56 |
| | MESM [20] | 0.73 | 0.08 | 0.59 | 0.23 | 0.06 | −0.11 | 0.12 | 0.06 | −0.19 | −0.28 | −0.23 | −0.17 | 0.07 |
| | CWM [7] | 0.36 | 0.06 | 0.67 | 0.34 | 0.23 | 0.18 | 0.23 | 0.18 | 0.37 | 0.13 | 0.49 | 0.23 | 0.29 |
| | Proposed | 0.91 | 0.08 | 0.87 | 0.46 | 0.57 | 0.34 | 0.22 | 0.25 | 0.46 | 0.08 | 0.34 | 0.59 | 0.43 |

The DMOS values of the proposed method and the CWM are in the superior group. The proposed method outperforms, especially in Arial and Boat, which contain many textures. Consequently, the proposed method produces superior perceptual performances for various test images, compared to existing methods.

Figure 7 compares the images enhanced by each method. The images enhanced by the frequency domain methods appear to be more sharpened, compared to those by the spatial domain methods. With increasing the sharpness, the proposed method also embeds granular signals onto the texture signals. Therefore, the images enhanced by the proposed method have the most resolved detail signals. Similarly, in the DMOS test, the subjects selected the image enhanced by the proposed method as being the highest resolution image.

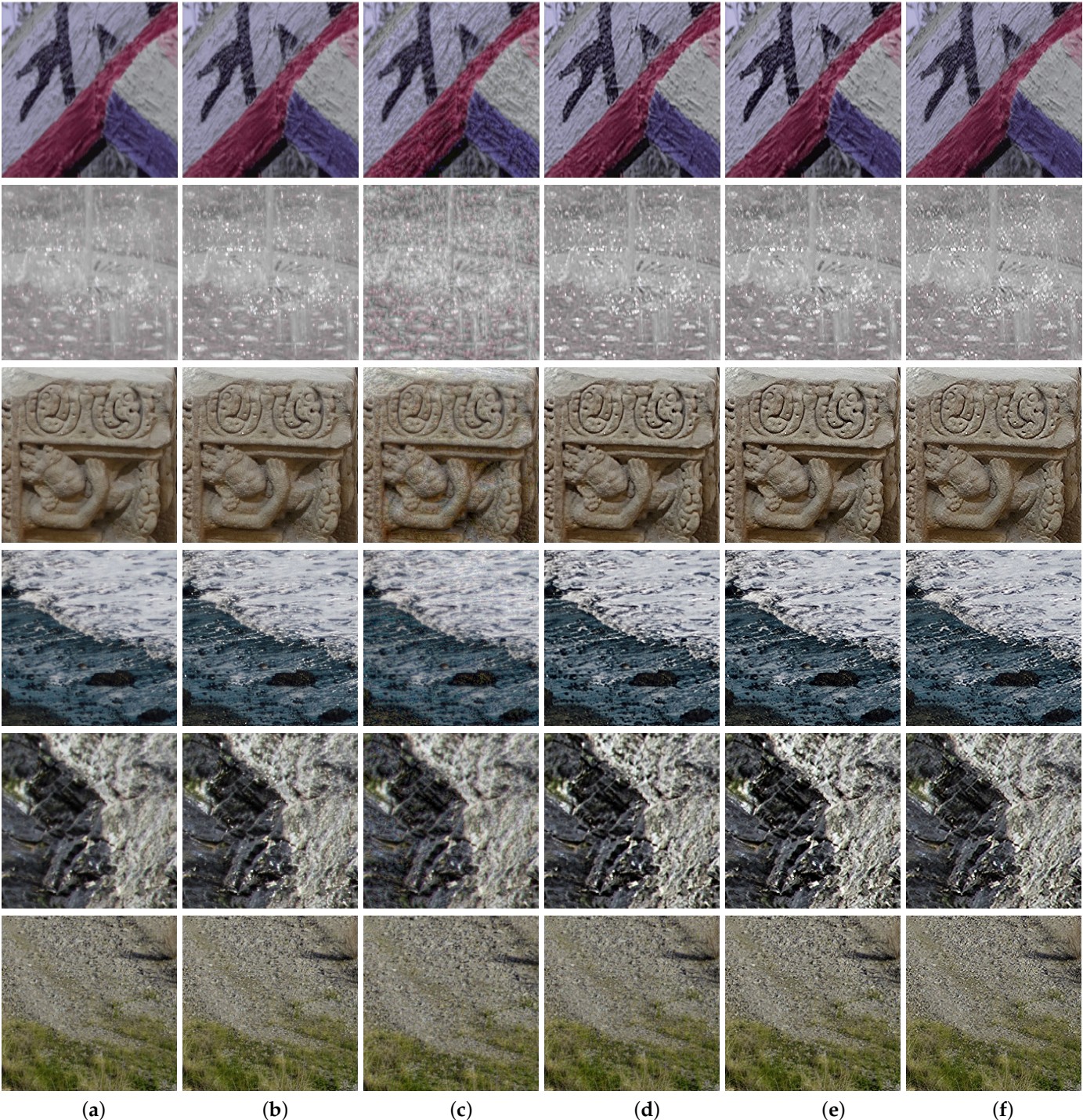

|     (a)     |     (b)     |     (c)     |     (d)     |     (e)     |     (f)     |

**Figure 7.** Enhancement results of images. (**a**) Original image. (**b**) By the Unsharp masking method. (**c**) By the CAIDE method. (**d**) By the MESM method. (**e**) By the CWM method. (**f**) By the proposed method.

Figure 8 analyzes how each method enhances detail image signals. For a better understanding, the horizontal signals of the enhanced images are presented in one dimension.

As shown in the rectangles, the frequency domain methods intensify locally variant signals compared to the spatial domain methods, demonstrating how the frequency domain methods produce better sharpness. The signals of the in circles indicate the granular signals embedded at locally variant signals. Such granular signals make the textures more finely resolved, allowing images to be perceived to have a higher resolution by humans. As shown by the circles in Figure 8e,f, the proposed method and CWM mainly generate such signals. CWM simply emphasizes higher frequency components over the entire image, and thus, it may generate granular signals that appear as noise in plain images. However, as discussed in Section 3, because the proposed method increases the high-frequency components according to $R(u,v)$, the proposed method does not produce noisy granular signals in plain images.

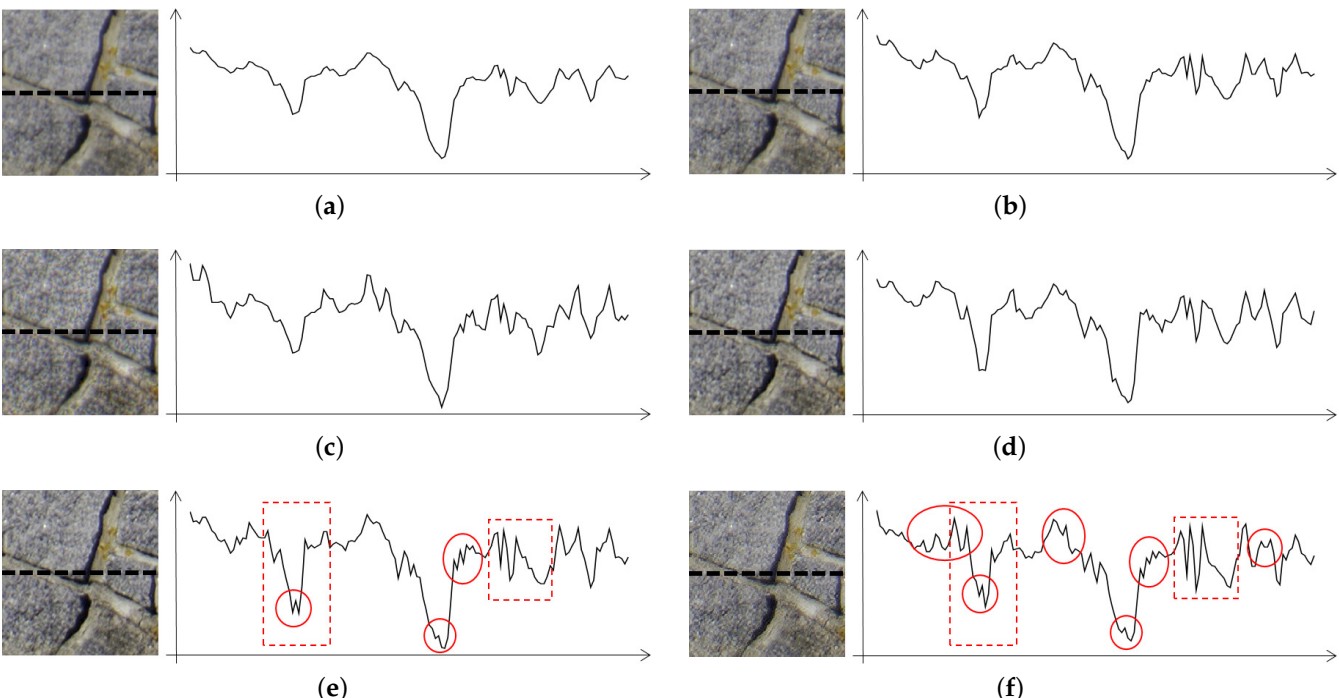

**Figure 8.** Comparison of enhanced detail image signal. (**a**) Original detail signal. (**b**) By the Unsharp masking method. (**c**) By the CAIDE method. (**d**) By the MESM method. (**e**) By the CWM method. (**f**) By the proposed method.

### 4.2. Artifact Analysis

The major artifacts related to the frequency domain methods are ringing artifacts and noise boost-up. Figure 9 compares the enhanced results of the plain, edge, and texture regions. In texture regions, such as grass areas, the proposed method and MESM enhance the texture image signals. However, in plain regions, such as cloud areas, whereas MESM produces noisy signals with a mosaic pattern because it increases the frequency energy without observing the image signal features, the proposed method does not produce any visible noise signals. This indicates that, through the energy scalar $R(u,v)$, the proposed method properly controls the enhancement effect in accordance with the underlying images.

In edge regions, such as wall boundaries, MESM produces ringing patterns along the edges because it enhances the image signals in all directions, including the direction perpendicular to the edge. However, the proposed method adjusts the enhancement direction to be parallel to the edge direction and rarely generates the ringing artifact.

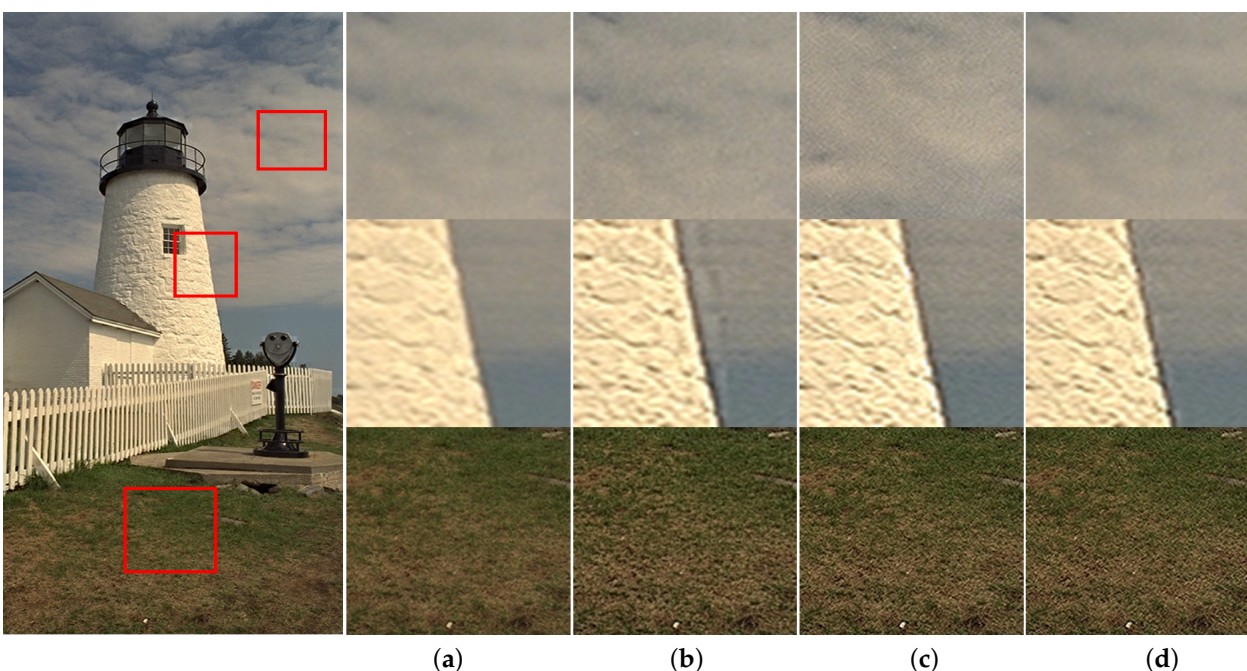

        (**a**)          (**b**)          (**c**)          (**d**)

**Figure 9.** Comparison of enhancement results at plane, edge and texture. (**a**) The original images. (**b**) By the MESM method. (**c**) By the CWM method. (**d**) By the proposed method.

## 5. Conclusions

We exploited the human perceptual contrast that measures the sensitivity of human visual perception to frequency components. Based on the perceptual contrast measure, we developed a frequency energy scaling-up method that not only emphasizes the local variant signals, but also strengthens the granular signals embedded in the local variant signals. Additionally, we developed a method to control the enhancement strength in adaptation to the characteristics of the underlying image signals. To reduce the ringing artifact, we devised a method that adjusts the enhancement direction to be parallel to the signal direction in the DCT domain. Therefore, the developed method enhances images to be perceived as clear and finely resolved, while avoiding any visible artifacts.

For improvements and further application of the proposed method, future work could proceed as follows. To increase enhancement performance, it is required to tune lambda adaptively to the underlying image signals. It would also be beneficial to extend the proposed method to color channels. For applications, in super-resolution, low-light, de-haze, and under-water areas, the parts—especially $R(u, v)$—of the proposed method could be positively exploited to efficiently enhance the detail images. Additionally, the proposed method could be applied to next generation displays for virtual and augmented reality [29,30].

**Author Contributions:** Conceptualization, S.B. and W.K.; methodology, S.B. and W.K.; software, S.B.; data curation, S.B.; writing—original draft preparation, S.B.; writing—review and editing, W.K.; supervision, W.K.; funding acquisition, W.K. All authors have read and agreed to the published version of the manuscript.

**Funding:** This work was supported by the BK21 plus program "AgeTech-Service Convergence Major" through National Research Foundation (NRF) funded by the Ministry of Education of Korea (5120200313836).

**Data Availability Statement:** The code is available on https://github.com/BangSeangbae/ImageEnhancement/tree/python (accessed on 10 October 2021).

**Conflicts of Interest:** The authors declare no conflict of interest.

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
