# Peer review of "DCT Domain Detail Image Enhancement for More Resolved Images"

_electronics, doi:10.3390/electronics10202461_

Round 1
Reviewer 1 Report
This paper proposes a method that not only emphasizes the local variant signals by scaling up the frequency energy in accordance with human visual perception but also strengths up the granular signals by embedding the alpha-rooting enhanced frequency components. Experimental results demonstrate the effectiveness of the proposed method.
First, the authors mentioned that "when an image signal is directed more vertically, the energy of the DCT coefficients is more condensed in the first row. Conversely, more horizontally directed signals have more energy than the DCT coefficients in the first column. " It would be better to show the phenomenon using a figure or an example.
Second, for a plane image, do we still need to enhance spectrums for plane images? Maybe smooth a plane image is better than adding spectrums? Such as the image denoising task.
Third, how about removing the artifact reduction path after the detail image enhancement path?
Finally, similar works of [a] and [b] could be referred to in this paper, which also tries to enhance image details.
[a] Single Image Dehazing via Multi-scale Convolutional Neural Networks with Holistic Edges
[b] Low-light image enhancement via a deep hybrid network
Reviewer 2 Report
In this paper, a novel image enhancement method in DCT-domain is proposed, where both local signals and the granular signal therein are emphasized. Experimental results have demonstrated the efficiency of proposed frequency energy scaling-up method, it is also remarkable that this work is human perception oriented. Following comments are recommended for further improving this paper.
- For a better comprehension, some concepts are suggested to be provided with more explicit illustrations, such as the ringing artifacts, and how to understand the image visibility, etc.
- Learning-based image-enhancement methods have been mentioned, which also witnessed many successful applications in relevant areas. Combined with traditional signal processing techniques in spatial or frequency domain, what are their merits, how can we define such work in a more in-depth way?
- Based on the expression of DCT coefficient F(u, v), it seems that u and v is totally equivalent, that is, F(u, v) = F(v, u). Hence this coefficient in a DCT block will be symmetric, I wonder what the inherent physical significance is.
- Notice many mathematical expressions have been used, make sure that each symbol is attached with an annotation.
- It is interesting that proposed method is human perceptual oriented, which is measured by the perceptual contrast (PC). Will this indicator possess a high generalization ability? Moreover, in the so-called subjective evaluation, to how much extent will the subjects influence the final estimation?
- As for the mentioned artifacts, except for the comparison presented in section 4.2, it is also suggested to conduct an ablation study to further demonstrate the effectiveness of designed signal direction adaptive enhancement strategy.
Reviewer 3 Report
This paper proposed an image detail enhancement for high-resolution displays. since the content seems technical sound but many technical and writing issues need to be filled. It is recommended to ask the author to rewrite the article before submitting it.
please provide more evidence and example and a stronger report to support the opinion or observation in the introduction.
The literature review is insufficient, please provide more recent research in the field.
The paragraph arrangement of the introduction is not appropriate. For example, the description in Figure 1 should be placed later or in the next section. The literature review should be placed earlier.
Please reconfirm the cited documents are follow the journal's instructions, eg, Kou et al. [1] ...
Please review SOTA methods/models, include the area in super-resolution, compressed sensing.
Is this method can de-blur for image, if yes, can you apply methods to blurred image then compare it to the original (blur-free) image.
Line58-61, if take DCT over the entire image is a time cost, please refer to others not use an entire image but still DCT research.
As mention the Line59-60, how to compare the noise of the granular signal.
Line 62-73, learning model sure it takes a longer time for the training phase, please discuss the run time and the outcome.
formula after Line97 seems to miss some statement of the parameter Cu Cv, and Cp is not in the formula. Please recheck all the formulas. Please make sure that all formulas are correct and meaningful. And some formulas are missing numbers.
Near Line 98, "Huan and Peli" missing the citation.
Section 3.3 Outline of the proposed method should be placed at the very beginning of the method to make it easier for the reader to understand.
confused at Table 1, DIQA seems to output the DMOS, and subjective (human) validations should be given MOS, but MOS and DMOS both missing in the table.
In table 1, the 0.43 at Avg. means only less than 10 subjects feel signal enhanced. what if three subjects are given 3 points but the rest of the subjects are given 0 pint is it Avg = 0.45? Does this result really mean that this method is better? Such a conclusion is difficult to convince people that this study is rigorous and correct.
Reviewer 4 Report
The submitted manuscript, "DCT-Domain detail image enhancement for more resolved images", explores perceptually inspired methods that help to improve the image quality of images displayed on a high-resolution display. The topic of the work is timely and actively followed by both computer vision and graphics systems communities.
The key idea of this work is to improve upon frequencies (image gradients) such that the images are perceived with higher contrasts, therefore perceptually higher resolutions.
This work can consider citing these relevant works in the field:
- The end application of higher resolution images can help to improve next-generation displays. Here is a survey on next-generation displays:
Koulieris, George Alex, et al. "Near‐eye display and tracking technologies for virtual and augmented reality." Computer Graphics Forum. Vol. 38. No. 2. 2019.
- Specifically, foveal regions of foveated displays can take advantage of perceptual improvements:
Kim, Jonghyun, et al. "Foveated AR: dynamically-foveated augmented reality display." ACM Transactions on Graphics (TOG) 38.4 (2019): 1-15.
- There are recent works that even claim peripheral vision images do not have to be perfect. It can indeed be swapped with metamers:
Walton, David R., et al. "Beyond blur: real-time ventral metamers for foveated rendering." ACM Transactions on Graphics (TOG) 40.4 (2021): 1-14.
- There are also works on the detectability of temporal artefacts in visuals:
Mantiuk, Rafał K., et al. "FovVideoVDP: A visible difference predictor for wide field-of-view video." ACM Transactions on Graphics (TOG) 40.4 (2021): 1-19.
- Some displays solely rely on representing discrete cosine transforms:
Akşit, Kaan. "Patch scanning displays: spatiotemporal enhancement for displays." Optics Express 28.2 (2020): 2107-2121.
- Some works expand on colour selection in images. Your DCT approach works across all colours but maybe in the future, and you can expand to colour space to look into alternative means to enhance the appearance of images with a DCT basis:
Nguyen, Chuong H., Tobias Ritschel, and Hans-Peter Seidel. "Data-driven color manifolds." ACM Transactions on Graphics (TOG) 34.2 (2015): 1-9.
Beside all these application suggestion which in my view has to be discussed in future work/introduction/discussion/related work sections. There are some things that I think has to be revised:
- For example, a discussion is needed about the highest resolution that the human visual system can resolve. The commonly accepted resolution power of the human visual system is 20/20, which translates to 1 arcmin. Probably, it is also a good idea to provide references of visual sensitivity and absolute limits of vision:
Elliott, David B., K. C. Yang, and David Whitaker. "Visual acuity changes throughout adulthood in normal, healthy eyes: seeing beyond 6/6." Optometry and vision science: official publication of the American Academy of Optometry 72.3 (1995): 186-191.
- I understand that the DCT coefficients in this new approach are manipulated (multiplier lambda) to enhance images. However, equation 13 seems rather simplistic to me as I would expect lambda to be chosen in a content-adaptive way. Furthermore, I have a hard time believing that this enhancement can work across all the images. If this is a correct observation, the authors should expand on a discussion about the limitations of their approach and what type of image subsets would be a good application base.
- Looking at Figure 5, it is visible that some features are at higher contrast. Therefore, they are easily resolvable. Then comes the question, how can an image be better than ground truth? I wonder if it makes sense to intentionally blur an image with a small kernel and trying to enhance that blurred image and compare it against the ground truth. This way, I assume that you can have a comparison between the original versus your method. It can further motivate streaming applications, where you intentionally send x4 smaller resolution images and reconstruct the higher resolution image in the size of the original image using your method.
Overall, I think I can support this manuscript for acceptance. However, the condition that I would put in place is related to the items raised in this review. Therefore, please kindly go through this review and address these points to entirely support acceptance in the future.
Round 2
Reviewer 2 Report
This paper can now be accepted for publication.
Author Response
We appreciate a reviewer's efforts for our manuscript.
Author Response

(The authors gave the same response as above.)
